# Interaction Map: A Visualization Tool for Personalized Learning Based on Assessment Data

Eric Ho  and Minjeong Jeon *

Department of Education, University of California, Los Angeles (UCLA), Los Angeles, CA 90095, USA; ericmho@ucla.edu
* Correspondence: mjjeon@ucla.edu

**Abstract:** Personalized learning is the shaping of instruction to meet students' needs to support student learning and improve learning outcomes. While it has received increasing attention in education, limited resources are available to help educators properly leverage assessment data to foster personalized learning. Motivated by this need, we introduce a new visualization tool, the interaction map, to foster personalized learning based on assessment data. The interaction map approach is engineered by the latent space item response model, a recent development in assessment data-leveraging social network analysis methodologies. In the interaction map, students and test items are mapped into a two-dimensional geometric space, in which their distances tell us about the student's strengths and weaknesses with individual or groups of test items given their overall ability levels. Student profiles can be generated based on these distances to display individual student strengths and weaknesses. Finally, we introduce a user-friendly, free web-based software *IntMap* in which users can upload their own assessment data and view the customizable interaction map and student profiles based on settings that users can adjust. We illustrate the use of the software with an educational assessment example.

**Keywords:** latent space item response models; educational assessment; visualization tool; personalized learning; interaction map; individual profiles; *IntMap*



## 1. Introduction

Education agencies are prioritizing personalized learning—the shaping of instruction to meet students' needs—to support student learning and raise student outcomes [1]. There is indeed promising research that addressing the diverse needs of students can help improve student achievements [2]. However, educators require resources to help them use assessment data to foster personalized learning since assessment data can help stakeholders understand students' strengths and weaknesses [1,3]. One way to address this challenge is by "encouraging the development of data assessment tools that are more intuitive and include visualizations that clearly indicate what the data mean for instruction" [3] (p. 58).

An example of such a visualization is an interaction map (also called a latent space) created by the latent space model [4] based on item response data. In the latent space, items and respondents are visualized as points in a two-dimensional geometric space. The positions of the items and respondents are determined by the interactions of the items and respondents with one another. For instance, respondents who are more likely to answer certain items correctly, even after accounting for item easiness and respondent ability, are located closer to one another on the latent space. An extension of this approach would allow practitioners to upload item response data from educational assessments and see

- which items a given student struggles with
- which items tend to assess the same concepts
- which items are generally easier or harder
- which students generally need more support

Although "personalized learning" is defined very generally, we believe that the afore-mentioned benefits of this approach would provide educators with finer-grained information about specific areas that their students are struggling with, based on the response patterns of the items on the assessment. This would allow educators to personalize the learning for those students by reinforcing their learning in those areas. Additionally, this approach can help the educator understand whether the items on their assessment are assessing the appropriate areas, thereby allowing them to make edits to the assessment as needed and improving the usability of the assessment data.

Therefore, to meet this need for personalized learning, we propose innovative visualizations of these student competencies so educators can foster personalized learning to raise student achievement. To that end, we utilize the latent space item response model (LSIRM [4]) which can yield useful information about student competencies because it yields a low-dimensional metric space called an interaction map to visualize items and respondents on a two-dimensional geometric space.

Specifically, our objectives in this study are as follows:

- We provide a brief overview of the LSIRM and its advantages compared to other conventional approaches such as the Rasch model [5] along with a brief illustration.
- We introduce *IntMap*, a Shiny App, to implement the LSIRM approach for educational assessment. We describe how the bubble chart interaction map from *IntMap* and strength-weakness profiles can be used to inform educators and practitioners about individual students' learning.
- We conduct a simulation study to compare the performance of the LSIRT with that of the Rasch model in the presence of uncaptured item-by-person interactions. We then provide an empirical illustration of our proposed approach with a real data example.

## 2. Background

In this section, we provide an overview of conventional approaches to understanding student strengths and weaknesses for the purposes of furthering personalized learning, the LSIRM, and a short preview of our approach.

### 2.1. Conventional Approaches

A commonly used approach involves calculating the proportion of correct items answered by each student and then ordering the students by the proportion to support the students who answered the fewest number of items correctly. While easy to calculate, this approach does not provide the educator with much information about domain-specific competencies for each student. For example, the teacher does not know whether a student who answered many items incorrectly is struggling with geometry items or not. The educator might then calculate proportions for each group of items that assess a specific domain, such as algebra or geometry. However, this is more complicated and presupposes that the items are adequately mapped to the domains as designed.

Additionally, this approach assumes items are of equal difficulty when this is almost certainly not the case. The educator could, in the vein of classical test theory, calculate the proportion of respondents who answered correctly for each item, but this runs into the circular dependency problem [6].

A more savvy educator may opt for traditional item response theory (IRT) models. One such model is the Rasch model [5]. The Rasch model is specified as

$$log \frac{P(X_{k,i} = 1 | \theta_k, \beta_i)}{1 - P(X_{k,i} = 1 | \theta_k, \beta_i)} = \theta_k + \beta_i,$$
(1)

where $\theta_k \sim N(0, \tau^2)$ is the ability of respondent $k$ and $\beta_i$ represents how easily item $i$ can be answered.

The ability estimates from the traditional Rasch model could yield proficiency estimates for each student, while also taking into account item difficulty. However, that

proficiency estimate alone does not disclose the specific areas a student needs help with. In fact, two students with similar ability estimates may indeed struggle with different topics. In that case, they might opt for a latent class model or diagnostic classification model, which can signal mastery or non-mastery in domains for a student. However, these again need to be specified a priori, and it is not always immediately clear that the test design holds in practice.

For example, two low-scoring students in the data, with IDs 1402 and 1155, have answered 23% and 10% of the items in the assessment correctly, respectively. If we estimate their overall ability using the Rasch model, we obtain their estimates of $\theta$ as $-1.09$ and $-1.81$. However, we know nothing about their strengths and weaknesses across the test items. All we know is that their overall abilities are low, but that is not enough to personalize learning for them. It is possible to look at their response patterns, but with over 600 respondents in the dataset, that can be very difficult.

The Rasch model also rests on a few assumptions. In particular, the model assumes that conditional on respondent abilities and item easiness, within-respondent and between-respondent responses to items are independent of one another. Most importantly, the model assumes that respondents with the same ability levels are equally likely to answer a given item correctly and that a respondent is equally likely to answer correctly two items of the same difficulty. These assumptions are unlikely to be met in practice, for instance, with culturally biased test items that require not only the overall ability to answer correctly but also certain cultural backgrounds that are not accounted for by the Rasch model.

### 2.2. The LSIRM Approach

In contrast, the LSIRM approach addresses all of the above challenges, as we will show in the later section. With this approach, the educator can still see the overall proficiency estimate as provided by the Rasch model (with less bias, as we will show in our simulation study later). However, the educator can also see the specific items that a student is struggling with, allowing them to identify the domains that they need more support with. Additionally, the LSIRM naturally clusters items that are similar to each other without requiring a priori knowledge. Therefore, the educator can see whether certain items designed to assess algebra concepts actually seem to be similar to one another. This approach provides not only general and domain-specific information about student proficiencies but also functions as an assessment validation tool. We will revisit this example and the results attained from the LSIRM approach in a later section, but first, we provide a background of the LSIRM.

### 2.3. Latent Space Item Response Modeling

In the latent space item response model (LSIRM [4]), both items and respondents have positions in a two-dimensional latent Euclidean space. Items and respondents that are closer in space are more affiliated than those that are separated by large distances. The larger the distance between a respondent and an item in the latent space is, the lower the probability of a correct response.

Suppose that respondent $k$ and item $i$ have positions $z_k \in \mathbb{R}^2$ and $w_i \in \mathbb{R}^2$ in the same latent, Euclidean space $\mathbb{R}^2$, respectively. Conditional on all positions of respondents $k$ and items $i$ in $\mathbb{R}^2$, the responses are independent and the log odds of a correct response $X_{k,i}$ is a function of the distance between the position of the respondent and the item as well as the item and respondent characteristics ($\beta_i$ and $\theta_k$, respectively):

$$log \frac{P(X_{k,i} = 1|\theta_k, \beta_i, z_k, w_i)}{1 - P(X_{k,i} = 1|\theta_k, \beta_i, z_k, w_i)} = \theta_k + \beta_i - ||z_k - w_i||_2, \qquad (2)$$

where $||z_k - w_i||_2$ is the Euclidean distance between the position of respondent $k$ and the position of item $i$. $\theta_k \sim N(0, \tau^2)$ is the person intercept and $\beta_i$ represents the item intercept, which can, respectively, be interpreted as the respondent's ability and item easiness after controlling for their distances in the latent space.

One can consider the distance term to be an interaction between the respondent and the item that is not explained by the person's ability and item easiness parameters. An important observation is that while respondents may have similar overall abilities represented by $\theta_k$, they may answer the same item correctly with different probabilities. Such a phenomenon indicates a violation of the conditional independence assumption, which is a strong but required assumption by most IRT models. Ample evidence is available to support that the conditional independence assumption does not hold in real performance assessment data [7–9].

The LSIRM weakens the restrictive assumptions of the Rasch model. The conditional independence assumption of the Rasch model states that item responses are independent conditional on the respondent and item attributes. Under the LSIRM, the conditional independence assumption is weakened such that the responses are conditional on the respondent and item attributes *and* the positions of the respondents and items in the interaction map.

It is important to note that the distance term, which represents item-by-person interactions given the person and item parameters, encapsulates the strengths and weaknesses that account for the difference in correct response probabilities among respondents with the same ability levels. As such, the LSIRM also weakens the homogeneity assumption of the Rasch model, enabling respondents of the same ability to answer the same item correctly with different probabilities and a respondent to answer two items of the same difficulty with different correct response probabilities. Our idea is to visualize these distances in a two-dimensional geometric space, which we call the "interaction map", to provide educators with finer-grained information about students' weaknesses and strengths, besides the more general information provided by the person and item intercepts.

### 2.4. Illustration

To compare the utility of our approach to existing approaches, we present a motivating example that comes from a math assessment for German 4th graders. The items are classified into various domains such as algebra and geometry. There are 664 students in the dataset assessed on 30 items. This dataset comes from loading `data.math` from the `sirt` package [10]. We present an example of an interaction map in Figure 1 based on this math assessment dataset. Each of the 30 items, marked as correct or incorrect for each respondent, is classified into domains (such as algebra, measurement, or geometry) and subdomains (such as addition, subtraction, or multiplication). The 664 respondents are represented by the smaller dots while the 30 items are represented by the bubbles. In the map, respondents who are closer to items are more likely to answer those items correctly even after accounting for respondents' overall ability and item easiness. Similarly, respondents farther from certain items generally struggle with those items, allowing educators to identify items or concepts that those students struggle with and to provide the appropriate interventions. Finally, the items are grouped by their similarities in the concepts they measure. The three-item groups seen in Figure 1 reflect different types of items designed in the assessment (algebra, geometry, measurement). Note that any information about items or respondents was not used in estimating the map. The item grouping was empirically shown, supporting the original test design. The bubble chart will be further explained in the next section.

Note that this may seem similar to Wright maps [11] in which items and respondents are placed on a continuum based on their easiness and abilities parameters, respectively. While there are similarities, it is important to note that the Wright map can only distinguish items and respondents based on those parameters while the LSIRM can distinguish items of equivalent difficulties and respondents of equivalent abilities [12]. We can see the distinctions between respondents of equivalent abilities in the student profiles presented later. Additionally, in a two-dimensional interaction map as shown here, greater distinctions can be seen in comparison to the one-dimensional Wright map.

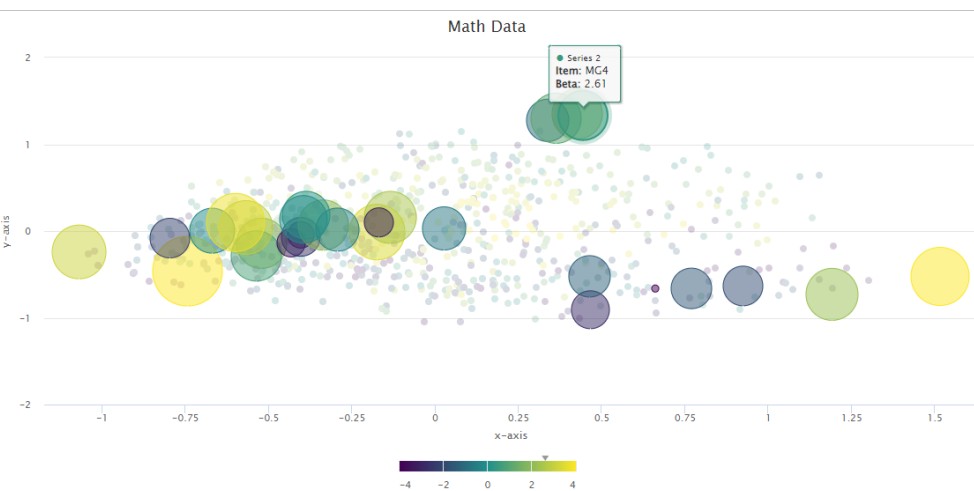

**Figure 1.** Bubble map of the math assessment dataset for German 4th graders. The 664 respondents are represented by the smaller dots while the 30 items are represented by the bubbles. Both dots and bubbles are colored by their respondent ability estimates and item easiness estimates, respectively, with a legend on the bottom denoting the values (lighter values for higher values, or easier items and higher-performing respondents). Additionally, only the bubbles are sized by their easiness, with larger bubbles signifying larger items. That way, item information is conveyed in two different, visually contrasting ways. Note that the interaction map is a tool to represent item-by-person interactions present in the data. Thus, the axes do not have specific interpretations and only serve to convey the scale of the distances among the items and respondents.

## 3. Methods

We present our methods for extending the LSIRM. We first show the application that allows users to upload their own assessment data and see their interaction maps and profiles. We then delve into the bubble chart, an improvement of the interaction map, and the student profiles.

### 3.1. IntMap: Interaction Map and Strength Profiles for Educational Assessment

The interaction map approach described in Section 2.3 can help practitioners foster personalized learning for two reasons: First, the interaction map and student profiles can be generated from assessment data. This is important because teachers generally prioritize the use of such data from formative and interim classroom assessments to inform their teaching [13–15]. Second, while there exist other data visualizations for assessment data, the visualizations from the interaction map provide finer-grained, unique item-level information that educators are generally interested in [16].

However, the full potential of the interaction map approach for educational assessment has not been fully shown yet in the literature. Plus, the estimation of the original model requires a complex Bayesian estimation procedure that may not be accessible to educators, practitioners, and applied researchers.

To fill this gap, we have developed *IntMap*, an application based on the `shiny` package [17] in R. This free-of-charge, web-based program is based on a user-friendly point-and-click system. The *IntMap* ShinyApp produces two types of visualizations, the interaction map and individual profiles, based on the input assessment data. The user interface uses the `shinydashboard` package [18] to create visually appealing dashboards within a Shiny application. In the application, multiple tabs allow users to upload their data, modify run settings (such as the burn-in period), display the interaction map, and download student profiles and estimated parameters. More advanced users can view convergence plots of the various estimated parameters. There are generally no limits to the size of data that can be uploaded to use this application. Practitioners can feel free to upload small-scale or large-scale assessment data (although computational times may be longer, especially if users choose a longer burn-in period or more iterations). The *IntMap* ShinyApp can

be found at https://ohrice.shinyapps.io/LatentSpace/ (accessed on 19 October 2023), a picture of which can be seen in Figure 2.

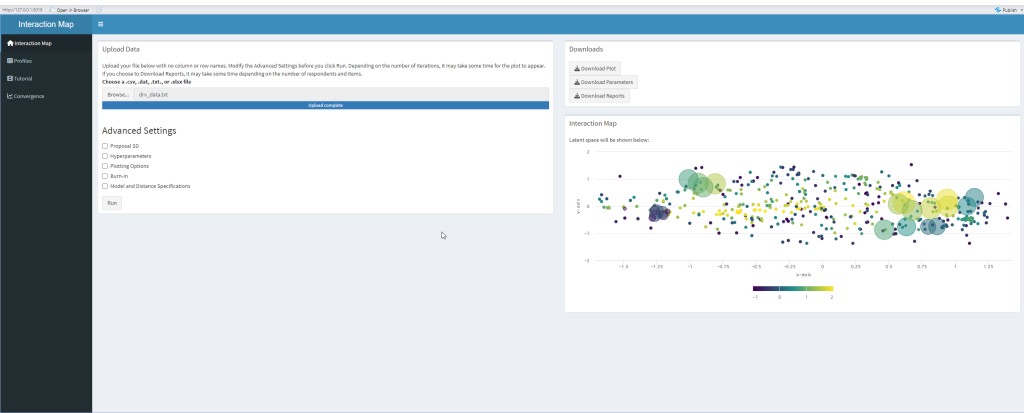

**Figure 2.** Screenshot of the Shiny application after a dataset has been processed. The interaction map can be seen on the right-hand side of the screen. There are buttons above the interaction map which allow the user to download various files.

In the following subsections, we will illustrate the visualizations, interaction maps, and individual profiles that *IntMap* offers, along with their interpretations for educational assessments.

### 3.2. Bubble Chart Interaction Map

*IntMap* produces a bubble chart-based interaction map containing information about the respondent and item parameter estimates ($\theta_k$ and $\beta_i$, respectively) in the visualization. This version of the interaction map also improves the interactivity of the visualization. As a reminder, Figure 1 displays an illustration of the bubble chart interaction map.

In the map, each respondent is represented by a point, and each item is represented by a bubble. The points and bubbles are colored by their ability or easiness parameter estimates, respectively. Darker points signify respondents of lower overall ability, while darker bubbles signify items with lower easiness parameters (or more difficult items). Additionally, the item bubbles are also sized according to the item easiness parameters, so larger bubbles signify easier items and vice versa.

Hovering over the respondents or items creates a tooltip that displays important information, such as the respondent/item identifiers and the respondent and item main effects. In contrast to the original interaction map, it is possible to visualize the main effects of the respondents and items while retaining the identifiers.

In general, this bubble chart interaction map answers questions important to educators without requiring them to delve into complicated statistics. First, they may wonder "Which items are generally easier/harder?" or "Which students are overall struggling and need more support?" These are common questions that they may ask. The bubble chart colors the respondents based on their levels of $\theta$, allowing educators to quickly identify such students. Similarly, the items are sized and colored by their easiness parameters. Note that this information is not readily available from the original interaction map presented by Jeon et al. [4]. Second, educators who want an in-depth analysis may ask, "Which items are certain students struggling with?" Again, the distances from respondents to items are apparent, allowing educators to see not only which students are struggling but also which items. An educator can zoom into a darker dot in the lower left-hand corner of the map to learn not only that particular student is struggling but also that they are having trouble with items in the top-right quadrant of the map.

In the interaction map shown in Figure 1, note that the items are naturally grouped according to the concepts they assess. The top group consists of items prefixed by "MG", which are items related to the "MG" testlet and measure the domain of "Measurement".

The bottom left group of items are prefixed by "MA", "MB", "MC", "MD", "ME", and "MF". "MA" to "MD" items all belong to the arithmetic domain. "ME" and "MF" items belong to the measurement domain, although they belong to a different subdomain from the "MG" items. Finally, the bottom right items are prefixed by "MI" and "MH" which all belong to the geometry domain.

This chart shows us that there are meaningful differences among the items depending on the domains they measure. By gauging the distances from the respondents to each group of items, we can try to determine which items certain students struggle with.

It would also be helpful to see which students are overall struggling before focusing on the specific items they need more support with. The interactivity of the bubble map allows users to more easily view information about overlapping items. The color-coded dots also allow us to see the overall abilities of students.

The bubble map shows us that the "measurement" items in the top group are relatively homogeneous in difficulty since they have roughly the same colors and size. In contrast, the other two item groups are relatively more heterogeneous in difficulty—some items are markedly easier than others. This may or may not be intended, but the educator cannot only see that the items are differentiated by the domains they assess but also the respective overall difficulty of each item. For the educator, this can be a mark of confidence in the assessment, as the items are differentiated by the different math domains they assess. For the test designer, the map can help validate their design.

One observation is that in the interaction map, respondents of very high abilities and very low abilities tend to be equidistant from the item groups since those students are equally likely (or unlikely) to answer all items correctly. In the original interaction map, both groups of students tend to be clustered in the center of the latent space. The strength of the bubble map is that users can distinguish between the two groups of students by looking at the colors of the respondents' points.

Finally, the advantage of the bubble chart is that it can help users sift through cluttered interaction maps. In the original interaction map, the respondent and item identifiers can be overlaid, especially when there are a lot of respondents and items in the dataset, making it hard for the user to distinguish between them. The tooltip allows users to carefully sift through respondents or items that would otherwise be cluttered together and distinguish them. As such, there are no set rules on the ideal number of respondents or items to use in this bubble chart.

### 3.3. Individual Strength-Weakness Profiles

Educators may wish to understand individual student performance alongside general trends. For this purpose, *IntMap* produces individual profiles created from the distances between an individual student's position and the positions of all the items on the interaction map. These distances can represent specific respondent-item interactions—for example, the greater the distance between the student and an item, the less likely the student is able to answer the item correctly. These distances can be interpreted as measures of student weaknesses or strengths that go beyond the conventional measures of respondent abilities provided by standard IRT models.

We visualize these profiles as bar charts in *IntMap*, a visualization technique that shows a relationship between a part and a whole or compares categories [19]. We use the bar charts to represent personalized learning by showing a student's strengths or weaknesses to individual test items given their overall ability and item difficulty. In the bar chart, the height of the bar represents the distance from a respondent to that item.

As an example, Figure 3 shows the profiles of two students. Although each student has roughly the same value of $\theta$ or overall ability, we can see from the profiles that they are weak in different items. Respondent 1155 has higher bars for all items except the "MH" and "MI" items, suggesting that this respondent needs more support on the domains of arithmetic and measurement. In contrast, respondent 1402 struggles with the "MH" and "MI" items, suggesting that they could use more support with geometry concepts. These

profiles allow the educator to pinpoint specific areas that the students need more support with. Only looking at the $\theta$ by itself obscures these respondent-item interactions; therein lies the advantage of the map and profiles in revealing these important interactions.

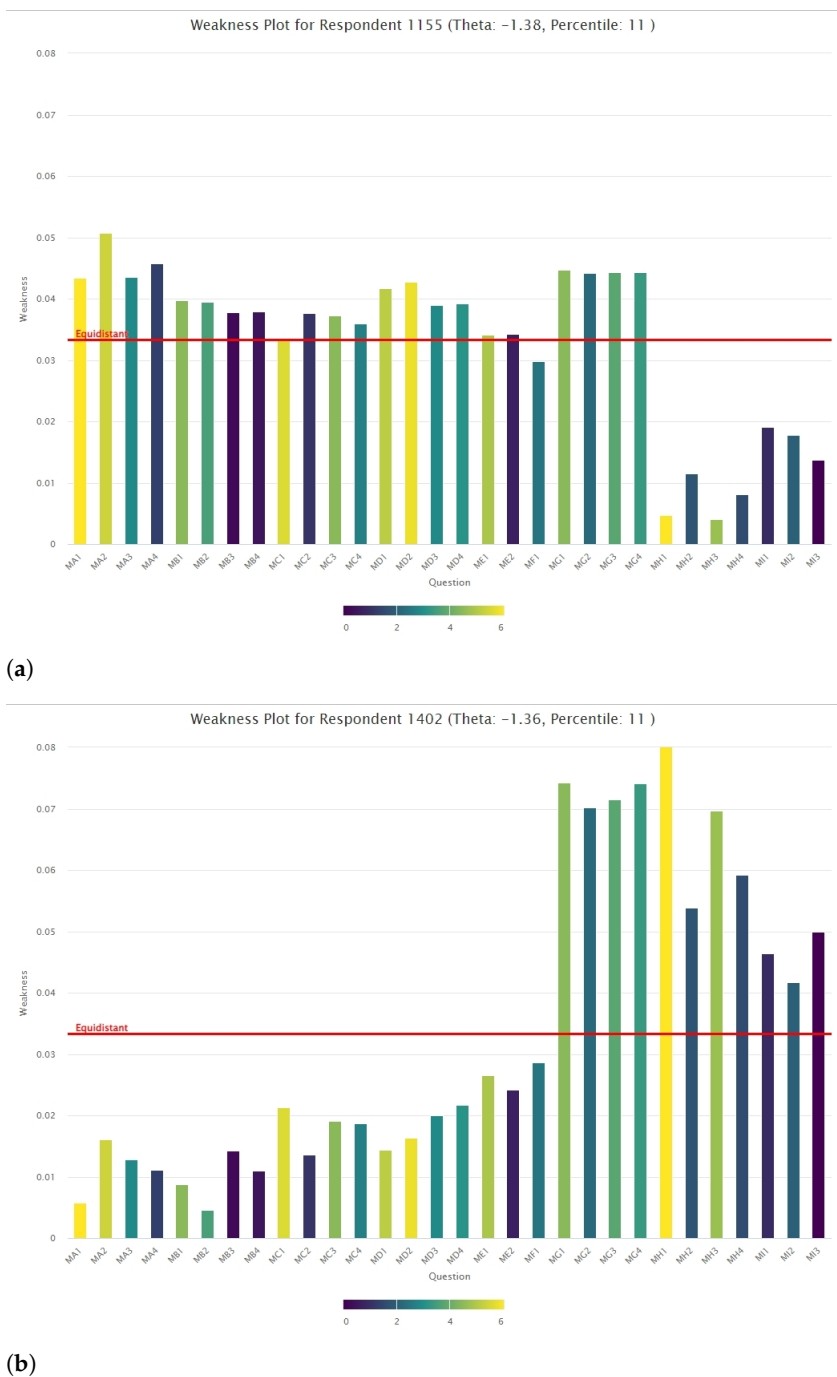

(**a**)

(**b**)

**Figure 3.** Presented here are profiles of two respondents in the mathematics assessment dataset from the `sirt` package. Each bar corresponds to an item, and the height of each bar corresponds to the distance from the respondent to that item. A higher bar, for example, represents a larger distance from the respondent to that item, suggesting greater "weakness" on that item. The equidistant line is a reference line to show what the height of the bar would be if the student were equally likely to answer all the items correctly or incorrectly. The respondents' ability estimates and percentiles relative to the rest of the sample are shown in the title. Each bar is colored by the easiness parameter of the corresponding item; the legend is shown below, with lighter bars signifying easier items. (**a**) Profile of respondent 1155; (**b**) Profile of respondent 1402.

The main takeaway for the educator is that the items appear to be differentiated by the algebra, geometry, and measurement domains. Those are three discrete skills that appear to be covered by this assessment. The educator can then delve into the interaction map and profiles to see which groups of students need support with which combination of the three domains.

## 4. Results

In this section, we present two sets of results—firstly, results from a simulation study comparing the performance of the LSRIM with that of the Rasch model, and secondly, results from applications of the aforementioned bubble charts and student profiles to real-life data. Both show the utility of our approach.

### 4.1. Added Values of the LSIRM: A Simulation Study

In the section above, we have illustrated the visualizations supplied by the LSIRM approach for educational assessment, which is a unique advantage of the LSIRM approach. The LSIRM approach offers an additional benefit, by reducing bias in ability estimates in the presence of uncaptured item-by-person interactions. Here we present simulation study results that demonstrate this point.

### 4.1.1. Data Generation

To simulate local dependence for a subset of the respondents, we consider a scenario in which respondents run out of time toward the end of the test and omit responses for those items. Omitted responses are then treated as incorrect responses; in this case, local dependence occurs between the respondents and the items with omitted responses. Standard IRT models do not capture such dependence [20].

In this setting, we consider N = 100, 200, 300, and 500 respondents and I = 20, 30, and 60 items, where responses are scored correct or incorrect. We first generated item responses from the Rasch model:

$$log \frac{P(X_{k,i} = 1|\theta_k, \beta_i)}{1 - P(X_{k,i} = 1|\theta_k, \beta_i)} = \theta_k + \beta_i, \tag{3}$$

where the item parameters and person parameters are generated from $\beta_i \sim Unif(-3, 0)$ and $\theta_k \sim N(0, 2)$. These distributions were chosen based on the distributions used in the simulation study presented in Appendix A in Jeon et al. [4]. We used those distributions because, in preliminary analyses, we wished to compare the simulated data generated in Appendix A and the data generated under our method. Using similar distributions allows us to make this comparison.

We then generated omitted responses by considering the responses of the first 50 respondents (regardless of their ability levels) to the last 10 items and setting them to be incorrect. This introduces the dependencies induced by the omitted items to respondents of varied abilities, not just to lower-performing respondents, allowing us to see that the interaction map from the LSIRM can indeed distinguish such individuals regardless of their abilities. While it is more realistic to have all of the lower-performing students omit the items, doing so would confound our results—perhaps those students were clustered by their overall abilities and not by the dependencies introduced by the omitted items. Since the sample size and the number of items are different in each data condition, the proportions of the respondents and the test items with omitted responses vary across the conditions (see Table 1). We generate 50 replicated datasets for each of the 12 conditions. We then fit the Rasch model and the LSIRM to each simulated dataset, extracted the person ability estimates, and compared the results.

**Table 1.** N is the number of respondents and I is the number of items. The entries indicate the proportion of respondents (first value) and test items (second value) with omitted responses at the end of the test. For example, 0.1/0.16 indicates that 10% of respondents showed omitted responses to the last 16% of the test items at the end of the test.

|  | I = 20 | I = 30 | I = 60 |
|---|---|---|---|
| N = 100 | 0.5/0.5 | 0.5/0.33 | 0.5/0.16 |
| N = 200 | 0.25/0.5 | 0.25/0.33 | 0.25/0.16 |
| N = 300 | 0.16/0.5 | 0.15/0.33 | 0.16/0.16 |
| N = 500 | 0.1/0.5 | 0.10/0.33 | 0.1/0.16 |

### 4.1.2. Results of the Simulation Study

In Figure 4, we present the distribution of mean errors as boxplots. Recall that there are 50 replicated datasets. The person-specific error (i.e., the difference between the estimated $\theta$ and the true value) was calculated for each person. These errors were averaged over all respondents in the replicated dataset to yield the mean error, for a total of 50 mean errors (one mean error for each of the 50 replications).

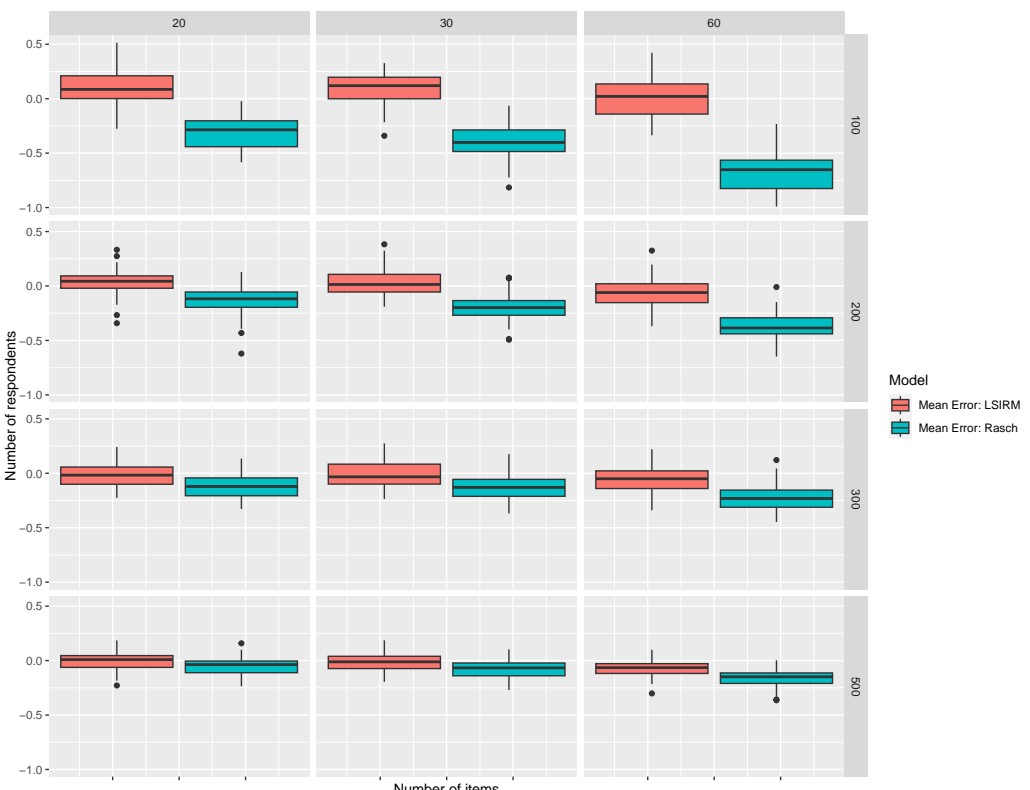

**Figure 4.** Distribution of mean errors across 50 replications. A mean error is calculated for each replication. Red indicates the results from the LSIRM and blue indicates the results from the Rasch models. The numbers on the top represent the number of items while the numbers on the right-hand side represent the number of respondents.

The mean errors are computed for all respondents, including both those who answered the last ten items on the assessment incorrectly and everyone else. It is shown that the magnitude of the mean error is less (closer to zero) when the respondent abilities are estimated using the LSIRM compared to when they are estimated using the Rasch model. Generally, the bias is more pronounced when there are fewer respondents although the pattern seems to hold for all the cases. The mean errors for each condition can be found in Table A1.

In conclusion, we have replicated a real-life scenario in which local dependence may occur and shown that in estimating the true respondent abilities, the LSIRM outperforms the Rasch model. This is important for practitioners who would no doubt prefer to receive more accurate estimates of their students' overall abilities to better understand which students would require more support in general. While the Rasch model may be easier to estimate, we believe that this is an important advantage of the LSIRM in delivering more accurate respondent abilities in the presence of local dependencies. Although practitioners may opt for the Rasch model first for preliminary estimates, the model selection process described in Jeon et al. [4] may be useful in ascertaining whether item-by-person interactions are present in the data being analyzed, i.e., whether the LSIRM is worthwhile to use for the given dataset.

### 4.2. Real Data Applications: Problems in Elementary Probability Theory

In this section, we demonstrate the usefulness of the bubble chart and profiles using responses from an assessment of elementary probability theory. There are a total of 504 respondents and 24 items (12 items presented in the first part and 12 items presented in the second part). The students are first given instruction about how to calculate the classic probability of an event (pb), the probability of the complement of an event (cp), of the union of two disjoint events (un), and of two independent events (id). Data were collected by Pasquale Anselmi and Florian Wickelmaier at the Department of Psychology, University of Tuebingen, in February and March 2010. This dataset can be loaded from `probability` in the `pks` package [21] in R along with additional information such as the exact wording of the items.

Results of the Real Data Application

In this analysis, we only consider the 12 problems shown in the first part. Based on Figure 5, it seems like there is one cluster of items in the bottom lefthand corner, consisting of items b104, b110, b111, and b112. One interesting thing to note is that those items all deal with assessing students' understanding of independent events. Therefore, it seems that these items are mainly differentiated by whether they assess computing the probability of independent events. The educator can decide for themselves whether this test is working as intended. Either way, the educator can ascertain that student responses can be differentiated by whether they understand how to calculate the probability of independent events.

Figure 5 shows that those items generally tend to be more difficult than the others.

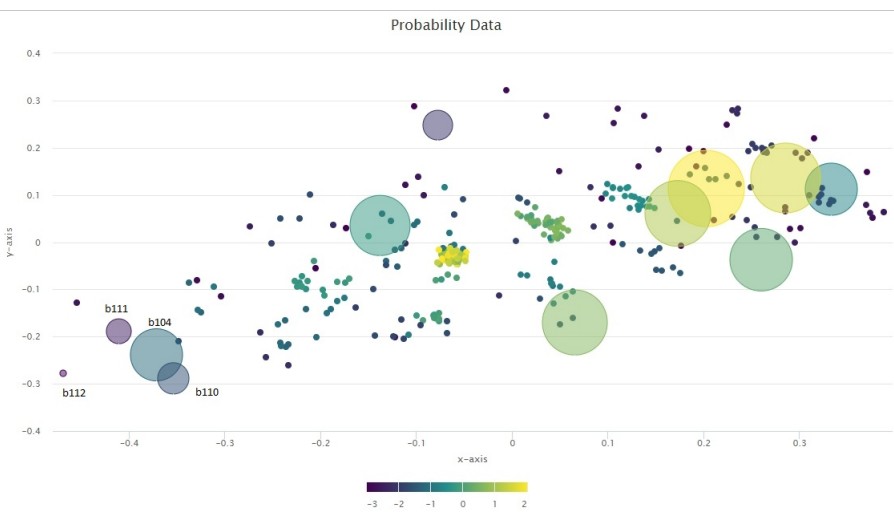

**Figure 5.** Bubble map of the probability assessment dataset. The 504 respondents are represented by the smaller dots while the 12 items are represented by the bubbles.

Finally, Figure 6 shows the profiles of two students, each with similar values of the person ability parameter. However, respondent 45 shows greater aptitude for answering items b104, b110, b111, and b112 correctly. This instructor may understand that this respondent does not have difficulty computing probabilities based on independent events but needs more support with other items. On the other hand, respondent 157 needs more support in understanding how to compute the probability of independent events. It is worth emphasizing again that because of these profiles, we can note these two respondents of similar abilities answer the same set of items (which have the same easiness parameters) with different correct response probabilities.

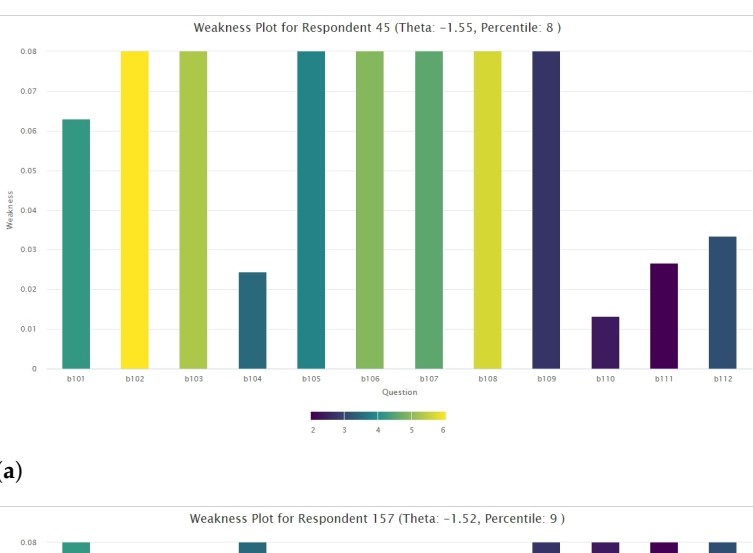

(**a**)

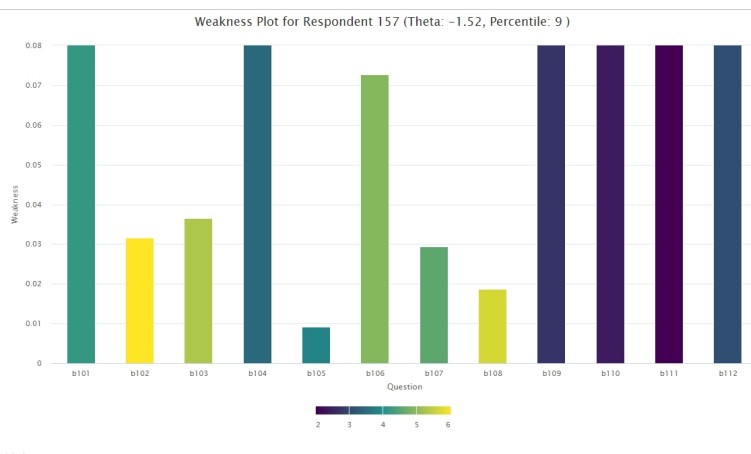

(**b**)

**Figure 6.** Presented here are the profiles of two respondents in the probability assessment dataset. These respondents have similar overall ability estimates yet they exhibit different strengths and weaknesses. For example, respondent 157 seems to struggle more with items related to independent events. (**a**) Profile of respondent 45; (**b**) Profile of respondent 157.

The main takeaway is that being able to compute the probability of independent events appears to be a discrete skill that students may or may not have trouble with, depending on their position in the latent space. We were able to see this by first looking at the bubble chart for item clusters. Then, the educator can look at profiles to see which students need more support with this specific skill based on their distances to items belonging to that cluster. The interaction map and profiles are best used in conjunction.

## 5. Discussion

In this paper, we have shown that the LSIRM is an innovative method with attractive psychometric properties, such as the ability to detect dependencies, allowing for less biased

estimates of respondent abilities. The LSIRM overcomes many of the limitations posed by traditional IRT models and other methods in fostering personalized learning. For example, while the Wright maps only show relationships between the main effects of persons and items, the interaction map shows the interaction effects given the main effects, thereby allowing one to distinguish between items of equivalent difficulties and respondents of equivalent abilities (something the Wright map cannot do). Additionally, based on our simulation study, there is evidence to suggest that the LSIRM provides less biased estimates of student ability compared to those provided by the Rasch model under a certain definition of local dependency. Most importantly, the interaction map and profiles can yield valuable insights for educational practitioners to foster personalized learning. The interaction map can provide a high-level view of student strengths and weaknesses while also providing diagnostic information about assessment items. The profiles allow practitioners to take a deeper dive into the abilities of a specific student. The utility of the interaction map and the profiles has been shown using datasets from different educational assessments.

To tie together the interaction maps and profiles, we presented *IntMap*, a Shiny application we have developed to improve the utility of the LSIRM approach for educational assessment. *IntMap* supplies visualizations of interaction maps and individual profiles that provide not only information that educators generally desire and have attained in the past (such as overall respondent ability and item easiness) but also specific information about the items that respondents struggle with.

## 6. Conclusions

*IntMap* can help educators provide the appropriate interventions to support the learning of their students. Perhaps most importantly, the use of these tools does not require advanced statistical knowledge. Researchers and psychometricians may also find the *IntMap* applications useful since they deliver details of the MCMC used to compute the item and respondent positions. It is our hope that the K–12 teacher—who may administer smaller, formative assessments—and the college professor—who may oversee larger classes—could find this tool useful; the former may find the student profiles useful for formative purposes while the latter may use the bubble charts to sift through the potentially cluttered interaction map due to the larger number of respondents and items.

As mentioned by the U.S. Department of Education [3], the development of intuitive and visually appealing data assessment tools can help educators leverage their assessment data to enhance their instruction and further personalized learning. Recently, novel visualizations and applications similar to those suggested by the Department of Education have been developed to further the goals of educators in supporting student success. For example, Bowers et al. [22] have developed visualizations that chart the longitudinal trajectories of students throughout high school to help policymakers and educators support student success. We hope that our contribution of *IntMap* is part of this nascent movement to develop applications and visualizations, based on sophisticated and rigorous methodologies, which nevertheless are accessible and provide a wealth of information.

Future studies can improve the practicality of the LSIRM. On the modeling side, the LSIRM could ideally be extended to accommodate polytomous responses rather than just binary responses on assessments. It would also be elucidating to further investigate the psychometric properties of the LSIRM. In our simulation study, we have only presented one possible definition of local dependency, derived from the phenomenon of students not having enough time to complete the assessment in its entirety. Additional investigations of the improvement in bias under different definitions of local dependency would be warranted along with modifications to our simulation scheme, such as introducing varied numbers of omitted items instead of a constant number of omitted items. On the computational side, further investigations may help improve the speed with which the *IntMap* can provide interaction maps and profiles. Users with large-scale assessment data may experience slower runtimes with the application given that the MCMC sampler would need more time to process greater numbers of items and respondents. The Rasch model

may still be faster to estimate, although we hope that future work will deliver LSIRM parameter estimates faster. Additionally, it would be worth investigating the performance of the LSIRM with smaller sample sizes or numbers of items. Our preliminary analyses suggest that our approach does not require more items or larger sample sizes than the Rasch model. We have seen promising results with datasets with 30 respondents and five items. Jeon et al. [4] showed promising results from a dataset with only seven items, so we are hopeful that our tool will be similarly useful with small sample sizes and numbers of items. Regardless, it would be illuminating for future studies to investigate the performance of the LSIRM under different sample size conditions, specifically with very large and very small numbers of items and respondents. Finally, the user interface of the *IntMap* software can be further refined to meet the needs of educators and other practitioners. For example, with large numbers of respondents and items, the interaction map can become crowded and difficult to interpret. The software could be modified to allow users to zoom into specific parts of the interaction map to make interpretation easier. Additionally, we could modify the interface to allow users to upload test specifications and design parameters, allowing them to more easily compare those with the model-derived item clusters and even see strengths and weaknesses in relation to those content clusters rather than in relation to individual items.

**Author Contributions:** Conceptualization, E.H. and M.J.; methodology, E.H. and M.J.; software, E.H.; validation, E.H. and M.J.; formal analysis, E.H.; investigation, E.H. and M.J.; resources, M.J.; data curation, E.H.; writing—original draft preparation, E.H.; writing—review and editing, E.H. and M.J.; visualization, E.H.; supervision, M.J.; project administration, M.J.; funding acquisition, M.J. All authors have read and agreed to the published version of the manuscript.

**Funding:** This research was funded by the Dissertation Year Fellowship from the UCLA Graduate Division.

**Institutional Review Board Statement:** Not applicable.

**Informed Consent Statement:** Not applicable.

**Data Availability Statement:** The datasets used in this paper are publicly available in several R packages as stated in the paper. The procedure to obtain the data is documented in the paper. The code used for the simulation study and in the Shiny application can be found at https://anonymous.4open.science/r/IntMap-Files-BF0E (accessed on 19 October 2023).

**Conflicts of Interest:** The authors declare no conflict of interest.

## Abbreviations

The following abbreviations are used in this manuscript:

LSIRM    Latent Space Item Response Model
IRT      Item Response Theory

## Appendix A

**Table A1.** Figure 4 shows the distribution of the mean errors across the 50 replications. This table shows the mean errors (the average over all 50 replications) for each condition.

| Respondents | Items | Mean Error: LSIRM | Mean Error: Rasch Model |
|---|---|---|---|
| 100 | 20 | 0.086 | −0.320 |
|  | 30 | 0.106 | −0.399 |
|  | 60 | 0.023 | −0.676 |
| 200 | 20 | 0.042 | −0.130 |
|  | 30 | 0.031 | −0.198 |
|  | 60 | −0.063 | −0.369 |

**Table A1.** *Cont.*

| Respondents | Items | Mean Error: LSIRM | Mean Error: Rasch Model |
|---|---|---|---|
| 300 | 20 | −0.017 | −0.115 |
| | 30 | −0.014 | −0.133 |
| | 60 | −0.057 | −0.226 |
| 500 | 20 | −0.008 | −0.055 |
| | 30 | −0.012 | −0.078 |
| | 60 | −0.070 | −0.159 |

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
