# Peer review of "Interaction Map: A Visualization Tool for Personalized Learning Based on Assessment Data"

_psych, doi:10.3390/psych5040076_

Round 1

Reviewer 1 Report

In the abstract I would recommend incorporating the main advantages of this software as if they were the conclusions of the study and eliminating the references, citations, from the text of the abstract. It is a summary of the paper presented.

In point 2.2. reference is made to the Rasch model without citing any document that can be consulted to learn more about it.

A section clearly stating the objective(s) of the research and the method followed is missing. It goes from one section to another without clearly delimiting the aspects of the research. A concrete theoretical framework in which both models, Rasch and LSIRM, are explained and then the objectives pursued in order to respond to the research and elaborate the discussion of the manuscript. The manuscript is complex to read because it does not follow a linear argument that is compatible with the standards of a scientific article. They should review the article carefully and eliminate what does not contribute anything to the manuscript and specify what is necessary (e.g., Rasch model).

In section 3.2. figure 1 should appear after the text and not before.

Figure 3 does not refer to the color of each bar to which it refers. In the same figure a color range appears with a numbering but it is not explained.

In figure 5 it is not clear which are the items referred to in the text (b104, b110, b111, and b112). Improve the figure.

The Discussion section is more like the Conclusions section. The authors do not make a discussion of the results obtained in comparison with other models or studies, for example. They offer the conclusions they have reached once the model they propose has been put into practice. They should elaborate a section for discussion or include one that addresses the discussion-conclusion. In addition, it would be interesting to present the limitations of the study, as well as its theoretical and practical implications.

Author Response

Dear Reviewer:

Thank you for taking the time to review our manuscript. We have attempted to address your concerns to the best of our abilities and hope that these changes will satisfy you. We have provided our responses to your comments point by point below.

In the abstract I would recommend incorporating the main advantages of this software as if they were the conclusions of the study and eliminating the references, citations, from the text of the abstract. It is a summary of the paper presented.

The references have been removed from the abstract, and the main advantages of this software (accessibility and customizability) have been included.

In point 2.2. reference is made to the Rasch model without citing any document that can be consulted to learn more about it.

 A reference has been included, along with a description of the independence assumptions that the model entails.

A section clearly stating the objective(s) of the research and the method followed is missing. It goes from one section to another without clearly delimiting the aspects of the research. A concrete theoretical framework in which both models, Rasch and LSIRM, are explained and then the objectives pursued in order to respond to the research and elaborate the discussion of the manuscript. The manuscript is complex to read because it does not follow a linear argument that is compatible with the standards of a scientific article. They should review the article carefully and eliminate what does not contribute anything to the manuscript and specify what is necessary (e.g., Rasch model).

Right after the introduction, we have included a section called “Objectives,” delineating the objectives of this manuscript and the roadmap to help clarify the structure. The descriptions of the Rasch model and the LSIRM are encapsulated in the “Background” section, with additional elaboration of the Rasch model specification and its assumptions. The online application, bubble map, and student profiles are now encapsulated in the “Methods” section, followed by our simulation study results and real-life data applications. We hope this makes the manuscript easier to follow.

In section 3.2. figure 1 should appear after the text and not before.

We have moved this figure after the text.

Figure 3 does not refer to the color of each bar to which it refers. In the same figure a color range appears with a numbering but it is not explained.

We have added an explanation in the caption explaining the legend and the colors of the bars, which correspond to the easiness parameters of the items.

In figure 5 it is not clear which are the items referred to in the text (b104, b110, b111, and b112). Improve the figure.

We have included identifiers on the figure showing the relevant items.

The Discussion section is more like the Conclusions section. The authors do not make a discussion of the results obtained in comparison with other models or studies, for example. They offer the conclusions they have reached once the model they propose has been put into practice. They should elaborate a section for discussion or include one that addresses the discussion-conclusion. In addition, it would be interesting to present the limitations of the study, as well as its theoretical and practical implications.

We changed the “Discussion” section to the “Conclusions” section. We have included a paragraph in the Discussion section discussing practical implications and further elaborated on the limitations of the study. Specifically, we discussed how this application may emerge as one of many others in delivering novel, detailed insights into educational and assessment data for practitioners and how this application may experience longer runtimes with much larger assessment datasets. Another limitation we have included is that the interaction map can be crowded with a lot of people and items and may be difficult to interpret. But in this case, one may zoom in on some of the areas or some of the people/items to ease the interpretation.

Please do not hesitate to let us know if you require additional information or clarification. Thank you again for your consideration of our manuscript.

Sincerely,

The authors

Reviewer 2 Report

I enjoyed the starting point of this manuscript: making assessment data easier to digest for educators/decision-makers via visualizations. Indeed, the insights derived from such tools can also be used for personalizing education for learners. However, I have some concerns regarding the utility of the visualization presented in the current study. I will explain them one by one below:

- I see that LSIRM is more accurate than the Rasch model in capturing local dependency between the items and respondents. However, if this is the only condition that makes LSIRM more accurate than Rasch, then I don't necessarily agree with the value of this approach over Rasch as Rasch is much easier to estimate.

- In my opinion, item-person maps (or Wright maps) can do pretty much the same job as the interaction map presented in the paper. We could add colors to differentiate difficulty and ability, as well as content groupings. It would not be identical to the interaction map, but it could easily convey the same message. 

- What are the ideal number of students and items for this tool to be used? I see how the visualization can potentially be used by educators, but I do not think that an educator would be willing to review data from 600 students presented on an interaction map. So, what are the parameters around the usability of this tool? After how many items would the tool be less useful (due to creating a more complex figure)? Maybe the interaction map is mostly for learners, rather than educators/instructors who interpret the data. 

- Where is this tool going to be useful? In a classroom setting in K-12 or a higher education classroom? Classroom assessments in K-12 typically involve much fewer items (especially formative ones) and students. So, I can't see how this tool could be useful for such settings. Maybe it is more suitable for large-size classes in higher education.

- What are the assumptions of LSIRM over Rasch? The authors presented the model as if it had no assumptions at all. The primary assumptions of this model must be explained clearly.

- On page 3, the authors explain that the theta is the person intercept. Is it similar to a random intercept estimated for the Rasch model within the GLMM framework? Are LSIRM and GLMM similar in that sense (in terms of parameterization)?

- In my opinion, the interaction map is not easy to interpret for several reasons. First of all, the x-axis and y-axis are not properly labeled. How could an educator know what these axes represent? On top of that, there are color and size layers for the bubbles. I think more descriptive notes on these elements and maybe a legend for the figure is necessary.

- After reviewing the individual strength-weakness profile plot, I thought plotting the difference between the difficulty level of each item (based on Rasch) and the person's ability could give the same type of message. This would be centered around the probability of 0.5, which is a bit easier to understand for someone who is not familiar with these models. I think the individual profile plot can be modified further to highlight weaknesses more explicitly while demonstrating the strengths in relation to the content clusters. Each learner should be able to see the item clusters based on the test design, not the clusters derived by the model.

-  In the simulation study, why is the variance for ability 2? The authors did not necessarily select a typical standard normal distribution.

- How did the authors select respondents with omitted items? Randomly? We know from the literature that mostly low-performing students tend to produce omitted items near the end of the test. The authors should clarify how they determined students with omitted items. Also, having the same number of omitted items across all of these students is a very unrealistic assumption.

- Overall, I ask the authors to clarify where this approach could be more or less useful, with a clear take-away message for readers and practitioners who may use their tool eventually.

Author Response

Dear Reviewer:

Thank you for taking the time to review our manuscript. We have attempted to address your concerns to the best of our abilities and hope that these changes will satisfy you. We have provided our responses to your comments point by point below.

- I see that LSIRM is more accurate than the Rasch model in capturing local dependency between the items and respondents. However, if this is the only condition that makes LSIRM more accurate than Rasch, then I don't necessarily agree with the value of this approach over Rasch as Rasch is much easier to estimate.

Thank you for your comment. At the end of the simulation study, we clarified that we do not advocate for the wholesale abandonment of the Rasch model but encourage users to consider this approach, as the LSIRM provides potentially useful information about respondents and test items, which may not be obtainable from the Rasch model. As we mentioned in the revised manuscript, one can use the model selection process described in Jeon et al. (2021) to evaluate whether the LSIRM may be worth using (i.e., whether item-by-person interactions are present in the data worth investigating with the LSIRM). Our intention with the simulation study was to provide some evidence that one may be safe considering the LSIRM without worrying about the ability estimation being impacted by the approach. In addition, we also have included in the “Conclusions” section an elaboration of the limitations of this approach (e.g., the ease of estimation) compared with the Rasch model.

- In my opinion, item-person maps (or Wright maps) can do pretty much the same job as the interaction map presented in the paper. We could add colors to differentiate difficulty and ability, as well as content groupings. It would not be identical to the interaction map, but it could easily convey the same message.

Thank you for this comment. Wright maps distinguish items based on their difficulties and respondents on their abilities, but the LSIRM can distinguish between items of equivalent difficulties and respondents of equivalent abilities. That is, what is shown in the interaction map is residual relationships (or interaction effects given the main effects), whereas Wright maps show relationships between the main effects (of persons and items). This can be seen in our real-life data applications. We have clarified this point and a reference to a thesis elaborating on this feature in the manuscript, at the end of the first illustration of the interaction map approach and in the Conclusions section.

- What are the ideal number of students and items for this tool to be used? I see how the visualization can potentially be used by educators, but I do not think that an educator would be willing to review data from 600 students presented on an interaction map. So, what are the parameters around the usability of this tool? After how many items would the tool be less useful (due to creating a more complex figure)? Maybe the interaction map is mostly for learners, rather than educators/instructors who interpret the data.

There is no set rule for the number of students and items for this tool to be used (we leave the decision to use the tool to the discretion of the user), although we do note in the Conclusion section that too many students or items can create clutter in the interaction map. We do note that with the bubble map, even with large numbers of items and respondents, which could lead to clutter, the tooltips can allow users to sift through clusters of items and respondents to find specific respondents or items if they wish. We included an elaboration on this in the section regarding the bubble maps.

- Where is this tool going to be useful? In a classroom setting in K-12 or a higher education classroom? Classroom assessments in K-12 typically involve much fewer items (especially formative ones) and students. So, I can't see how this tool could be useful for such settings. Maybe it is more suitable for large-size classes in higher education.

The proposed approach does not require more items or larger sample sizes than the Rasch model. From our experiments, the proposed approach works well with as small as N=30 people and I=5 items. We did not present these results since it is not the main purpose of the current paper. In the Conclusions section, we discuss the need for further simulation studies to verify the minimum sample size conditions for the proposed approach but note that the LSIRM had been applied to datasets with small numbers of items. We believe both kinds of educators may find this tool useful, since the bubble chart can negate the clutter that may emerge from larger-sized classes in higher education, and the student profiles could be useful in the formative instances you have described in K-12 classrooms, as commented in the Conclusion section. 

- What are the assumptions of LSIRM over Rasch? The authors presented the model as if it had no assumptions at all. The primary assumptions of this model must be explained clearly.

The assumptions of the Rasch model are delineated under “Conventional Approaches.” We have noted the assumptions of the LSIRM in the description of the LSIRM.

- On page 3, the authors explain that the theta is the person intercept. Is it similar to a random intercept estimated for the Rasch model within the GLMM framework? Are LSIRM and GLMM similar in that sense (in terms of parameterization)?

Yes, they can be thought of similarly.

- In my opinion, the interaction map is not easy to interpret for several reasons. First of all, the x-axis and y-axis are not properly labeled. How could an educator know what these axes represent? On top of that, there are color and size layers for the bubbles. I think more descriptive notes on these elements and maybe a legend for the figure is necessary.

Thank you for this comment. We have included additional notes in the caption of Figure 1 describing the interpretations of the axes and the color and size layers.

- After reviewing the individual strength-weakness profile plot, I thought plotting the difference between the difficulty level of each item (based on Rasch) and the person's ability could give the same type of message. This would be centered around the probability of 0.5, which is a bit easier to understand for someone who is not familiar with these models. I think the individual profile plot can be modified further to highlight weaknesses more explicitly while demonstrating the strengths in relation to the content clusters. Each learner should be able to see the item clusters based on the test design, not the clusters derived by the model.

Thank you for your suggestions. The idea behind the student profiles is that even persons of the same abilities answering the same items may have different correct response probabilities since the Rasch model does not account for dependencies which induce these different probabilities. Our current profiles are able to convey this additional piece of information, while taking only the difference between the item difficulty level and the person's ability may not. For example, in Figure 6, we compare two respondents of similar abilities with regard to the same set of items. If we had only taken the difference between the difficulty levels of those items and their ability estimates, we would not have been able to discern the differences in their correct response probability.

We have noted that the interaction map should be used in conjunction with the profiles to demonstrate more clearly the weaknesses and strengths in relation to model-derived clusters. We have included your suggestion of seeing item clusters based on the test design in our areas of future work in this manuscript.

-  In the simulation study, why is the variance for ability 2? The authors did not necessarily select a typical standard normal distribution.

We chose those distributions based on the ones used in the simulation study presented in Appendix A in Jeon et al., (2021). We used those distributions because, as in Jeon et al. (2021), we also required the use of simulated data and, in preliminary analyses, wished to compare the simulated data generated in Appendix A and the data generated under our method. Using similar distributions allows us to make this comparison. We have included this context in our manuscript.

- How did the authors select respondents with omitted items? Randomly? We know from the literature that mostly low-performing students tend to produce omitted items near the end of the test. The authors should clarify how they determined students with omitted items. Also, having the same number of omitted items across all of these students is a very unrealistic assumption.

In the simulation study, we described how the first 50 respondents in the dataset were chosen to have the omitted items, regardless of their levels of ability. While you are correct in that mostly low-performing students would tend to produce the omitted items towards the end, in the preliminary exploratory analyses from our simulation study, we wanted to show that the interaction map can still cluster respondents with the omitted items, regardless of their ability levels. Having all of the lower-performing students omit the items, while more realistic, would confound our results - perhaps those students were clustered by their overall abilities and not by the dependencies introduced by the omitted items. These additional notes have been included in the “Data Generation” subsection. We agree that this does not reflect reality entirely and have noted this as a limitation in our Conclusions, but we maintain that we had a good rationale to do so for this specific purpose.

- Overall, I ask the authors to clarify where this approach could be more or less useful, with a clear take-away message for readers and practitioners who may use their tool eventually.

Thank you for your comments. We have described the advantages of our approach while also being honest with the limitations of the tool in the last section of our manuscript.

Please do not hesitate to let us know if you require additional information or clarification. Thank you again for your consideration of our manuscript.

Sincerely,

The authors

Round 2

Reviewer 1 Report

Dear authors, thank you very much for your answers to the questions and suggestions.

I have reviewed the article again and, although I see that there is a section dedicated to the objectives, as such, they are not well defined in this section. The objectives are described in the infinitive and are the elements that will allow us to carry out the research and discussion of the work.

Sections 4 and 5 would be part of what is known as Results. And they would not be put separately. In addition, the discussion has been eliminated when they should have kept the discussion where the objectives (which have not been correctly explained) that were set out before the method would be answered. The conclusions should be the main contributions of the study.

References do not appear at the end of the article

Author Response

Dear Reviewer 1,

Thank you for your comments.

I have reviewed the article again and, although I see that there is a section dedicated to the objectives, as such, they are not well defined in this section. The objectives are described in the infinitive and are the elements that will allow us to carry out the research and discussion of the work.

We removed the “Objectives” section and merged it into the Introduction to improve flow. We have also reformatted our objectives in the infinitive and in bullet points to improve clarity.

Sections 4 and 5 would be part of what is known as Results. And they would not be put separately. In addition, the discussion has been eliminated when they should have kept the discussion where the objectives (which have not been correctly explained) that were set out before the method would be answered. The conclusion should be the main contributions of the study.

We have merged Sections 4 and 5 into one section called “Results.” We have also re-included the Discussion section with the relevant text. 

References do not appear at the end of the article

We have modified the .tex file so that you should be able to view the references now.

Thank you again for your time and consideration.

The authors

Reviewer 2 Report

I still have one more question based on the following comment I made and the answer that the authors provided:

 In my opinion, the interaction map is not easy to interpret for several reasons. First of all, the x-axis and y-axis are not properly labeled. How could an educator know what these axes represent? On top of that, there are color and size layers for the bubbles. I think more descriptive notes on these elements and maybe a legend for the figure is necessary.

Thank you for this comment. We have included additional notes in the caption of Figure 1 describing the interpretations of the axes and the color and size layers. 

What I meant in my comment is that the tool itself should be updated, not necessarily the figure captions in the manuscript. It would be great if the authors added the elements I suggested to their tool before the tool became publicly available to other researchers or practitioners. 

Author Response

Dear Reviewer 2,

Thank you for your comment.

What I meant in my comment is that the tool itself should be updated, not necessarily the figure captions in the manuscript. It would be great if the authors added the elements I suggested to their tool before the tool became publicly available to other researchers or practitioners.

In the “Tutorial” tab of the Shiny application, we have added a text box describing various aspects of the interaction map and how the interaction map can be interpreted.

Thank you again for your time and consideration.

The authors

Round 3

Reviewer 1 Report

Dear authors, thank you very much for the answers to the questions raised. The latest version of the paper is much better although the objectives are not written in the infinitive. Please revise this because it is not well phrased and weakens the quality of the study.

Best regards.

Author Response

Dear Reviewer 1,

Thank you for your comments.

Dear authors, thank you very much for the answers to the questions raised. The latest version of the paper is much better although the objectives are not written in the infinitive. Please revise this because it is not well phrased and weakens the quality of the study.

We have been assured by the editor that the paper will undergo a thorough English

check after acceptance. We will make any necessary edits then.

Thank you for your time and consideration.

The authors